# Carbon Monoxide Production during Bio-Waste Composting under Different Temperature and Aeration Regimes

**DOI:** 10.3390/ma16134551

**Published:** 2023-06-23

**Authors:** Karolina Sobieraj, Sylwia Stegenta-Dąbrowska, Christian Zafiu, Erwin Binner, Andrzej Białowiec

**Affiliations:** 1Department of Applied Bioeconomy, Wrocław University of Environmental and Life Sciences, 37a Chełmońskiego Str., 51-630 Wrocław, Poland; karolina.sobieraj@upwr.edu.pl (K.S.); sylwia.stegenta-dabrowska@upwr.edu.pl (S.S.-D.); 2Department of Water-Atmosphere-Environment, Institute of Waste Management and Circularity, University of Natural Resources and Life Sciences, Muthgasse 107, 1190 Vienna, Austria; christian.zafiu@boku.ac.at (C.Z.); erwin.binner@boku.ac.at (E.B.); 3Department of Agricultural and Biosystems Engineering, Iowa State University, 605 Bissell Road, Ames, IA 50011, USA

**Keywords:** carbon monoxide dehydrogenase (CODH), bio-waste treatment, lab-scale composting, kinetics, oxygen deficit

## Abstract

Despite the development of biorefinery processes, the possibility of coupling the “conventional” composting process with the production of biochemicals is not taken into account. However, net carbon monoxide (CO) production has been observed during bio-waste composting. So far, O_2_ concentration and temperature have been identified as the main variables influencing CO formation. This study aimed to investigate CO net production during bio-waste composting under controlled laboratory conditions by varying aeration rates and temperatures. A series of composting processes was carried out in conditions ranging from mesophilic to thermophilic (T = 35, 45, 55, and 65 °C) and an aeration rate of 2.7, 3.4, 4.8, and 7.8 L·h^−1^. Based on the findings of this study, suggestions for the improvement of CO production throughout the composting process have been developed for the first time. The highest concentrations of CO in each thermal variant was achieved with an O_2_ deficit (aeration rate 2.7 L·h^−1^); additionally, CO levels increased with temperature, reaching ~300 ppm at 65 °C. The production of CO in mesophilic and thermophilic conditions draws attention to biological CO formation by microorganisms capable of producing the CODH enzyme. Further research on CO production efficiency in these thermal ranges is necessary with the characterization of the microbial community and analysis of the ability of the identified bacteria to produce the CODH enzyme and convert CO from CO_2_.

## 1. Introduction

The constantly growing problems related to the existing excessive use of raw materials and fossil fuels by humans, combined with excessive consumerism and waste generation, have resulted in an urgent need to find new ways to produce goods. An important role in this challenge is played by the use of secondary raw materials, still rich in useful compounds and substances, but directed imprudently to landfills or incineration plants. In this way, approaches that fit into the ideas of the circular economy and bioeconomy, based on constantly maintaining a high value of materials and products through their turnover in closed loops, are becoming more and more important. Continuously developed solutions together with a well-established system approach and the involvement of all relevant entities can result in an even more efficient use of valuable raw materials.

The area of interest of the bioeconomy includes organic materials, such as bio-waste, and green or agricultural waste. As substrates rich in organic carbon, they are currently mainly subjected to composting or anaerobic digestion processes; however, their potential is constantly recognized in other areas, such as biorefinery processes [1]. Despite the development of this field of study in recent years, so far, the possibility of coupling “conventional” waste-processing processes with the production of valuable biochemicals or biofuels by converting generated process gases has not been taken into account [2]. However, the bio-waste composting process, during which net carbon monoxide (CO) production has been observed, has the potential to couple these processes [3,4,5,6,7,8]. The benefits of using CO both as a product itself and as a substrate for subsequent conversions in the chemical industry have been extensively discussed in the literature [9,10,11,12]. Since the production of CO from biowaste is inevitable due to its high organic carbon content, the use of the generated CO appears to be an attractive solution to serve the objectives of the circular economy and the biorefinery approach. Although CO is commonly produced by the thermal gasification of bio-waste, this process requires energy to be supplied to dry the substrates [13]. Therefore, biowaste composting may become an economically and ecologically competitive method of obtaining CO. There are, however, no studies that specifically address this scenario to date.

So far, research on CO production from aerobic bio-waste processing is characterized by a high degree of uncertainty, largely based on sometimes contradictory reports on the optimal conditions for the formation of this gas. However, analyses conducted by a few researchers in the 1990s and early 2000s identified two main variables influencing CO formation: oxygen concentration and temperature [7,8,14,15,16]. Further studies conducted at the beginning of the second decade of the 21st century defined the production of CO from waste composting as a combination of a dual nature: abiotic and biotic processes [5,6,17,18].

CO production is stimulated by the increased availability of O_2_; higher concentrations of CO were recorded after turning the material into a compost pile due to the aeration of areas where anaerobic conditions had previously developed [7]. In addition to the positive correlation to O_2_ availability, CO generation is also temperature dependent. Based on studies conducted for soils, during which the soils produced CO during the day at a temperature of 30–40 °C and became a net CO sink at night (temperature < 30 °C), the researchers conjectured a hypothesis about the physicochemical sources of CO generation [19]. This hypothesis was later confirmed by analyses by Phillip et al. [6], where higher CO levels were found in sterilized samples of composts. An unambiguous indication of the nature of CO formation was, however, impossible due to the observed fluctuations in the concentration of this gas during waste composting. A high level of CO is characteristic of the initial phase of the process (a few hours [7,16] or even 10 min after starting the process, at a temperature of 35 °C [7]); then, the concentration of CO decreases and increases again after approx. 5–8 days (50 °C [3,7,16]). A similar trend was also observed during analyzes of CO production from wetlands [20]. Due to the increasing concentration of CO_2_ occurring in parallel with the lowering of CO concentration, the gradual depletion of CO is associated with its microbial oxidation [7]. Although the first rapid increase in CO concentration is explained by the thermochemical processes of its generation, e.g., the abiotic degradation of fatty acids, polyphenols, and aromatic acids, the next peak was defined as the biotic [8]. As reported by Haarstad et al. [8] and Rich and King [20], CO production during composting is linked to methanogenesis and the activity of methanogens since the strong peak of CO concentration reached even 2022 ppm (0.2%) at a very low level of O_2_.

An important aspect of adjusting the bio-waste composting process to CO production and subsequent coupling of this gas conversion in biorefinery processes is to increase the concentration of generated CO. Only finding the optimal conditions of the composting process for CO release will allow for an increase in its formation, which will translate into the technological application of these processes. Since it is known that a combination of abiotic and biotic processes (that can occur in parallel) are happening to stimulate or compete with each other, the issue of CO production during the bio-waste composting process should be treated holistically under the process conditions that are most conducive to its generation. This study aims to investigate CO production potential during bio-waste composting under controlled laboratory conditions by varying aeration rates and temperatures. For this purpose, a series of composting processes were carried out in conditions ranging from mesophilic to thermophilic (T = 35, 45, 55, and 65 °C) and aeration rates from 2.7 to 7.8 L·h^−1^. Daily measurements of CO concentration were used to determine the kinetic parameters of the decrease in CO concentration to find optimal conditions for its generation.

## 2. Materials and Methods

### 2.1. Materials

Bio-waste from the composting plant of Lobau, Vienna (Austria), collected from green and less densely populated areas of the city of Vienna, was used for composting on a laboratory scale. Waste material consisted of plant-based waste collected separately from bio-waste bins, i.e., vegetables and windfall fruit, leaves, tree and shrub cuttings, lawn clippings, and wilted flowers. The bio-waste was previously shredded, sifted, and screened in the mechanical treatment unit of the Lobau facility, and combined with chopped branches. Bio-waste samples were collected each time a new series of the composting process was started from a freshly formed waste pile (1–2 days old). A fresh waste sample of approx. 50 kg was collected manually with a shovel using the quartering method into plastic trays. After transporting them to the laboratory, they were again shredded to obtain a homogeneous particle size (elimination of larger pieces of wood blocking the material in the bioreactor). Substrate samples before the process were characterized (Section 2.5).

### 2.2. Bio-Waste Composting

For bio-waste composting on a laboratory scale, 12 adapted glass desiccators (bioreactors, Vienna, Austria) with a volume of ~7 L each were used. The working space of the bioreactor was divided into three areas: the upper part (headspace); the middle part, where bio-waste was placed (composting chamber); and the lower part, separated from the middle part by a perforated plastic screen for leachate collection (Figure 1). The bioreactors were equipped with covers with one closed valve and exhaust air outlet; after cooling, the exhaust air was collected in gas collection bags. A gas-tight connection of the cover with the body of the reactor was ensured by applying Vaseline on the edge of the bioreactor. A port with a screw cap in the middle part of the bioreactors allowed for the manual insertion of a thermocouple into the composted material to measure the temperature of the waste. Air supply was adjusted individually (from 2.7 to 7.8 L·h^−1^) for each bioreactor and was inserted at bottom part of the reactor by a hose connection and flow controller.

Bioreactors were weighted (accuracy 0.01 g, initial weight) and then placed in a climate chamber in rows of 4 bioreactors on one level (shelf, Figure 1b). To avoid the thermal ‘shelf effect’ (different ambient temperatures), variant repetitions in triplicates were analyzed vertically. Depending on the experiment variant, the climate chamber was set at 35, 45, 55, and 65 °C (Table 1). Since the CO concentration is highest in the initial phase of the process [3,4], one composting cycle lasted 14 days. After 7 days, the bio-waste was removed, manually mixed in a cuvette for aeration, and placed back in the bioreactor (compost turning).

### 2.3. Measurements of Process Gases Concentration and Temperature

CO concentration (ppm) measurements were conducted every 24 h with the first measurement 24 h after placing the bioreactors in the climatic chamber and starting the process. The Polytector III G999 gas concentration analyzer with a measuring range of 0–1000 CO ppm (GfG Gesellschaft für Gerätebau mbH, Dortmund, Germany) was manually connected to silicone tubes that were connected to the gas bags collecting the process air from each of the bioreactors separately. The measurement was carried out until the concentration values stabilized (approx. 1 min). Since CO concentration was reported as co-dependent with O_2_ and CO_2_ levels, O_2_ concentration (%) was measured in parallel using the same method and analyzer. After each CO and O_2_ measurement, the analyzer was disconnected for a short pause to return to ambient levels (CO~0 ppm, O_2_~20.2%, CO_2_ ~0%). CO_2_ concentration (%) was analyzed every 24 h by an infrared gas analyzer (type GMA 052, GfG Gesellschaft für Gerätebau mbH, Dortmund, Germany) connected to the gas bags (Figure 2). During the measurement series, the port with a screw cap in each bioreactor was opened, the thermocouple (Testo 925, Testo SE & Co. KGaA, Titisee-Neustadt, Germany) was inserted into the material (~15 cm depth) and the compost temperature was measured (±0.5 °C). On each level of the shelf, a 1 L plastic container filled with water was placed and the temperature was measured with a thermocouple (Testo 110, Testo SE & Co. KGaA, Titisee-Neustadt, Germany) to determine the ambient temperature (±0.2 °C).

### 2.4. Bio-Waste Samples Collection

After this process, each of the bioreactors was weighed (accuracy 0.1 g). Then, bio-waste from 3 bioreactors with the same aeration rate variant was placed in a plastic hutch and mixed manually with a metal shovel. The homogenized material’s properties were then analyzed (Section 2.5). Leachates from each bioreactor were collected separately with a pipette and placed into a plastic vessel and weighed (accuracy 0.01 g).

### 2.5. Bio-Waste and Compost Characterization

Substrates and compost samples were characterized according to methods published elsewhere [22,23]. Analyses of water content (WC), pH, electrical conductivity (EC), loss on ignition (LOI), total organic carbon (TOC), total nitrogen (TN), carbon/nitrogen ratio (C/N), respiration activity AT_4_, ammonium nitrogen (NH_4_-N) and nitrate nitrogen (NO_3_-N) content were carried out.

### 2.6. Calculations

The TOC, TN, C/N, NH_4_-N, and NO_3_-N content values of substrates and composts were calculated according to equations presented elsewhere [24].

The initial properties of the bio-waste and the final compost samples were compared to determine the process’s efficiency.

The kinetics of CO concentration decrease, during composting, were analyzed using linear and nonlinear least-squares regression. The models of the 0th-order (linear) and 1st-order (nonlinear) reactions were used for the analysis (Equations (1) and (2)):(1)$$C_{\mathrm{CO}}=a\cdot t$$
where:

C_CO_—CO concentration at time t, ppm;

a—regression coefficient indicating the decrease in CO concentration during the experiment, ppm·d^−1^);

t—time, days (d).
(2)$$C_{\mathrm{CO}}=C_{\mathrm{COmax}}\cdot e^{-k\cdot t}$$
where:

C_COmax_—maximum CO concentration, ppm;

k—decrease in CO concentration rate constant, days^−1^ (d^−1^);

t—time, days (d).

### 2.7. Statistical Analyses

All data were analyzed using Statistica StatSoft Inc., TIBCO Software Inc, and the analysis involved estimating the measurements’ mean, standard deviation, analyzing variance (CO concentration vs. process temperature and aeration), and correlation analysis (CO concentration vs. CO_2_, O_2_ concentrations, ambient temperature, and compost temperature for first 7 days of composting). Parametric tests (unequal-variance analysis and Tukey’s post hoc test at the significance level α = 0.05) were used to compare the differences between variants.

## 3. Results

### 3.1. Bio-Waste Initial Properties

The bio-waste, which was collected every ~2 weeks for the composting process, had similar TOC (33.0–36.9% d.m.), TN (1.3–1.6% d.m.) and C/N (23–28, Table 2) across the study period. The water content of the samples also did not differ much for substrates processed at 35 °C, 45 °C and 55 °C (~54%); only the bio-waste composted at 65 °C was characterized by higher moisture (61%). A similar trend was observed for organic matter content. The highest LOI was achieved by bio-waste processed at the highest temperature (71.0% d.m.), followed by substrates from the 35 °C variant (69.2% d.m.) and similar values for 45 °C and 55 °C samples (67.5 and 64.2% d.m., respectively). Bio-waste was acidic in most cases (pH equal to 5.6, 6.5, and 6.8 for 45 °C, 55 °C and 65 °C, respectively); the exception was waste composted at 35 °C, where the pH was 7.5. Electrical conductivity reached a similar level for all samples (~3 mS·cm^−1^), with the highest value exceeding 4 mS·cm^−1^ for substrates processed at 45 °C. The largest differences between the substrates were observed in the case of NH_4_-N and NO_3_-N content. The ammonium nitrogen content ranged from 140.46 to >800 mg·kg d.m.^−1^, with the highest values recorded for substrates at 45 °C and 55 °C (850.4 and 689.3 mg·kg d.m.^−1^, respectively). The NO_3_-N content was generally lower with a maximum of 169.6 mg·kg d.m.^−1^ (in the 65 °C variant).

The highest respiratory activity was found in substrates directed to the process at 65 °C (71.1 mg O_2_·g d.m.^−1^). Slightly lower, similar levels of AT_4_ were found for biowaste processed at 45 °C and 55 °C (~60 mg O_2_·g d.m.^−1^); a low index was observed for substrates in the variant with the lowest temperature (37.6 mg O_2_·g d.m.^−1^).

### 3.2. Composts Properties

When analyzing the properties of the composts after 2 weeks of the process, it was seen that the water content and organic matter content (LOI) reached a similar level for the samples composted within the same temperature, even if the aeration rate was different (Figure 3 and Figure 4). Generally, the highest values of both of these indicators were characteristic of the lowest aeration rate (2.7 L·h^−1^) except for the 65 °C variant, where LOI was higher for bio-waste aerated with 3.4 L·h^−1^. Similarly to the substrates, the TOC and TN values of the compost samples were similar for each temperature variant. TOC in composts was similar to that in substrates, while TN was slightly higher than in bio-waste samples (>30% d.m. and ~2% d.m. for TOC and TN, respectively, Figure 3 and Figure 4, Appendix A); hence, the C/N ratio for the composts decreased compared to the input values (<20). After the process, the pH of the bio-waste changed from acidic to alkaline; the only exception was compost processed at 45 °C with the lowest level of aeration (pH, in this case, was 5.3). As before, also after the process, the samples from the 35 °C temperature variant were characterized by the highest pH (>8.4). The highest EC was found for composts processed at 45 °C (ranging from 3.5 to 5.0 mS·cm^−1^ for the highest to lowest aeration variants, respectively, Appendix A). For the remaining temperature variants, the EC decreased compared to the values recorded for the substrates and did not exceed 3 mS·cm^−1^.

The largest variability between compost samples was noted for NH_4_-N and NO_3_-N content. In terms of temperature variants, the lowest NO_3_-N values were obtained by composts processed at 65 °C (from 3.3 to 25.2 mg·kg d.m.^−1^, Figure 3 and Figure 4, Appendix A), but when considering this indicator’s value with respect to the aeration variants, no clear trend was observed. The highest NO_3_-N content was characteristic of aeration of 3.4 L·h^−1^ at 35 °C and 55 °C (38.4 and 86.6 mg·kg d.m.^−1^, respectively), while for 45 °C and 65 °C, it was characteristic for bioreactors aerated with 4.8 L·h^−1^ (81.0 and 25.2 mg·kg d.m.^−1^, respectively). In turn, NH_4_-N content reached the lowest level for composts processed at the lowest temperature (<15 mg·kg d.m.^−1^); higher values were recorded for composts processed at 45 °C, 55 °C and 65 °C with maxima at aeration rates of 2.7 L·h^−1^ (1461.1, 390.9 and 265.4 mg·kg d.m.^−1^, respectively). These results indicate the most substantial nitrification in the case of the variant with the lowest temperature. This is consistent with observations reported in the literature. The optimal conditions for nitrification are the mesophilic temperatures (20–35 °C) and a pH from 7.5 to 8.0 [25,26], which was noted in the 35 °C variant in this experiment.

The lowest respiratory activity after 2 weeks of composting was characteristic of the material processed at 35 °C; samples from each aeration variant reached a similar AT_4_ value here (~13 mg O_2_·g d.m.^−1^). These composts can be considered stabilized since AT_4_ is <20 mg O_2_·g d.m.^−1^ [27]. For the other temperatures, only four samples did not exceed the required threshold (compost at 55 °C aerated with 3.4, 4.8, and 7.8 L·h^−1^ and at 65 °C with an aeration of 4.8 L·h^−1^). However, it is worth mentioning that the initial AT_4_ values for substrates depended on the thermal variant. In this way, bio-waste that was processed at 35 °C, with the lowest respiratory activity index, exhibited the lowest final value, while for bio-waste in variants at 45–65 °C, that initially reached an AT_4_ > 60 mg O_2_·g d.m.^−1^, the activity of microorganisms during the 14 days of the process did not decrease to the limit value.

Despite the use of different temperatures and aeration rates, each of the variants’ physical and chemical properties changed in ways typical to waste composting (Table 3). After 14 days of the process, the pH of the composts increased relative to the initial values of bio-waste; this increase, which was probably related to the degradation of organic and volatile fatty acids [28], ranged from approx. 11% at the highest temperature to >30% for material processed at 45 °C. An increase in the final values was also noted for TN. The highest increase was achieved at 35 °C (>38%), while the lowest at 45 °C (minimum 9.92%). This is consistent with the observations of other researchers who associated such a trend with the activity of nitrogen-fixing bacteria [29]. The remaining parameters were characterized by a decrease in value after 14 days of the process. For AT_4_, TOC and LOI, the decrease in the ranges from 32.91 to 85.02%, from 2.55 to 9.66 and from 5.15 to 11.70%, respectively, was related to the degradation of organic compounds by microorganisms; after metabolizing easily degradable compounds, the activity of microorganisms declined [30]. Due to the processes of mineralization and the gradual stabilization of waste [31], the C/N ratio was decreased from ~17% (the lowest aeration rate for temperatures of 45 °C, 55 °C and 65 °C) to >32% (in the variant with lowest temperature). Lower ECs for composts than substrates (from 0.49% to 27.25%) confirmed the gradual stabilization of processed waste [32]. Single exceptions to the general trend were noted for aeration rates of 2.7 and 3.4 L·h^−1^ at 35 °C (water content), 45 °C (pH, EC, water content and NH_4_-N content), 55 °C (NO_3_-N content) and 65 °C (NH_4_-N content, Table 3).

During the composting of biowaste, the largest total weight loss was observed for an aeration of 4.8 L·h^−1^ in each temperature variant (Table 4). With an increase in process temperature, mass loss increased (from 13.64% to 16.20% for temperatures of 35–65 °C). Moreover, mass loss in the form of leachate for each aeration rate was the highest at 55 °C. The more efficient the air supply, the larger the percentage weight loss in the leachate (16.47–20.10% for 2.7–7.8 L·h^−1^ aeration).

### 3.3. CO Concentrations

The average CO concentration reached the lowest values during composting at 35 °C, regardless of the aeration rate used (<100 ppm, Figure 5). In this thermal variant, CO also reached its minimum the fastest (for most repetitions in ~4–6 days of the process). As the process temperature increased, the CO concentration became less stable and the variations between daily measurements for repetitions increased (see Figure 5a,d). During bio-waste composting at 35 °C, the highest CO concentration was recorded for an aeration of 2.7 L·h^−1^ (up to 89 ppm); in this variant, these values reached the minimum the latest (day 9 of the process compared to day 5 for variants with aerations of 4.8 L·h^−1^ and 7.8 L·h^−1^, Figure 5a). The most stabilized CO concentration was observed when compost was aerated with 4.8 L·h^−1^ (the lowest variations).

Among all tested variants, the highest average initial CO concentration was observed in the composting process conducted at 45 °C with an aeration rate at 3.4 L·h^−1^ (>130 ppm, Figure 5b). However, in the case of the 2.7 L·h^−1^ and 7.8 L·h^−1^ aeration rate variants, the elevated CO concentrations (~40 ppm) were maintained until the 14th day of the process; for the highest aeration rate, these values were lower on the first day of composting, but remained at a slightly higher level on the last day (average 17 vs. 14 ppm CO for 7.8 L·h^−1^ and 2.7 L·h^−1^, respectively).

The highest average CO concentration on the first day of the composting process at 55 °C was characteristic of the variant with the lowest aeration rate, followed by variants with 7.8 L·h^−1^, 4.8 L·h^−1^ and 3.4 L·h^−1^ (190, 116, 106 and 35 ppm, respectively, Figure 5c). Compared to the process at 45 °C, the average CO concentration decreased faster, reaching several ppm in the second week. The largest differences between aeration rate variants were found during composting at 65 °C (Figure 5d). For the lowest aeration rate, the average CO concentration on the first day of the process was close to 300 ppm, while for the 4.8 L·h^−1^ variant, it was 3 ppm, and for 3.4 L·h^−1^ and 7.8 L·h^−1^, it did not exceed 100 ppm. The CO level was most stable at an aeration rate of 3.4 L·h^−1^; in the last three days of composting, it was 8–11 ppm, while for the experiment with the highest initial CO levels, it was 3–5 ppm (options 3.4 and 2.7 L·h^−1^, respectively).

This study confirmed that temperature and aeration level affect CO concentrations but only at low temperatures and aeration rates (35 °C and 2,7 L·h^−1^; Figure 6). Higher temperatures (>35 °C) and aeration rates (>3.4 L·h^−1^) did not influence CO production during the composting process (no statistically significant differences).

A statistically significant correlation between the CO concentration and all other investigated variables (concentrations of CO_2_ and O_2_ process gases, ambient and compost temperatures) for the first 7 days of the process was observed only in the case of the thermal variant at 35 °C (Table 5). CO concentration was inversely correlated with O_2_ concentration (Pearson correlation coefficient r = −0.47), while there was a positive correlation between CO level and compost temperature, ambient temperature, and CO_2_ concentration (strongest for the first, r = 0.6, r = 0.2 and r =0.4 for compost temperature, ambient temperature, and CO_2_ concentration, respectively). A positive correlation between CO and CO_2_ concentrations was characteristic only for this variant of the process. For the temperatures of 45 °C and 55 °C, there was a negative relationship between CO and CO_2_ levels, which was stronger for 55 °C (r = −0.6). Apart from the variant at 35 °C, the ambient temperature played a statistically significant role only at 65 °C; r was higher than that of the lowest temperature (0.3 vs. 0.2 for 65 °C and 35 °C, respectively).

### 3.4. CO Production Kinetics

For most cases, the decrease in CO concentration during composting was proceeded by the 1st-order reaction; only compost processed at 65 °C and aerated at 4.8 L·h^−1^ was consistent with the 0th-order reaction (Table 6).

Except for the 45 °C temperature variant, the trend for C_COmax_ was similar for the other thermal conditions: the highest values in the range of 185.3 ppm (35 °C) to 471.9 ppm (55 °C) were recorded for aeration rates of 2.7 L·h^−1^, while the lowest C_COmax_ was characteristic of an aeration of 4.8 L·h^−1^ (with a minimum of 32.5 ppm, Table 6, Appendix A). The highest average C_COmax_ among all aeration variants, equal to 244.4 ppm, was recorded for the 55 °C variant.

Similarly to the CO concentration and C_COmax_ values (characterized by higher deviations for successive temperature variants), the constant rate k within one thermal variant varied more with the increasing temperature of the composting process (Appendix A). However, in none of the analyzed cases did the constant k exceed 0.9 d^−1^. The most similar values between the different reactor aeration variants were observed for the temperature of 35 °C. Under these process conditions, the CO concentration decreased the fastest (k > 0.8 d^−1^). The lowest k value was recorded during composting at 65 °C with an aeration of 4.8 L·h^−1^ (k = 0.085 d^−1^), although the lowest average k was characteristic of the process carried out at 45 °C (k = 0.259 d^−1^). In addition, there was no trend in the reaction rate decrease in CO concentration between the aeration variants. The highest k was recorded for different aerations depending on the composting process temperature: 7.8 L·h^−1^ (35 °C variant and 65 °C, k = 0.850 and 0.670 d^−1^, respectively), 3.4 L·h^−1^ (45 °C variant, k = 0.453 d^−1^) and 2.7 L·h^−1^ (55 °C variant, k = 0.816 d^−1^).

The analysis of the average reaction rate indicated that the highest daily CO concentration for most thermal variants was achieved with the lowest aeration rate (the exception was the 45 °C variant, for which the coefficient reached the highest value of 114.2 ppm·d^−1^ with an aeration rate of 3.4 L·h^−1^). The average reaction rate ranged from >100 ppm·d^−1^ up to 478.8 ppm·d^−1^ (variant 55 °C, aeration 2.7 L·h^−1^).

## 4. Discussion

The conducted research proved that CO concentration varies depending on the temperature of the process and the level of aeration. Observations made for individual composting variants, however, highlight two CO production environments: at 35 °C and 65 °C, with simultaneous oxygen deficit.

Despite the forced, constant temperature level throughout the composting process, the phase of CO production reported earlier by the researchers was also noted during the experiment. For each of the analyzed thermal variants, the CO level was high at the beginning of the process, and then gradually decreased after about 7 days. This finding is consistent with the trend observed for composting various fractions of organic waste, including animal dung, leaves, grass, sewage sludge with bio-waste, green waste, and livestock waste [3,7,16]. The stimulation of CO production through the low aeration of composted waste was also consistent for all temperature variants, as it was also associated with the highest C_COmax_. Oxygen deficits were favorable for CO release at lower-than-optimal aeration (variant 2.7 L·h^−1^), indicating anaerobic processes as a probable source of CO production for each temperature variant. As mentioned earlier, Haarstad et al. [8] and Rich and King [20] came to similar conclusions in their research. During aerobic processing of municipal solid waste, the CO concentration even exceeded 2000 ppm, which the authors explained by the activity of methanogens at intermittent O_2_ loading [8]. In turn, a closer association with anaerobic biotic H_2_ generation was observed for the production of CO by wetland peats [20].

The intensification of net CO production associated with the presence of anaerobic conditions was also confirmed by the highest CO concentrations obtained during composting at 65 °C. This suggests a connection with the biological nature of CO formation, based on the activity of anaerobic microorganisms capable of producing the carbon monoxide dehydrogenase (CODH) enzyme. This enzyme catalyzes the reversible oxidation of CO to CO_2_ in the water–gas shift reaction [33] and thus it is responsible for both the production and conversion of CO. Potential microorganisms inhabiting the compost that produce oxygen-tolerant CODH are, e.g., *Desulfovibrio vulgaris* and *Carboxydothermus hydrogenoformans* [34]. In addition, this enzyme can be reactivated after a temporary occurrence of conditions with increased oxygen concentration [33,35]. Thus, turning the material after 7 days of the process could interrupt the biological production of CO, but would restart when O_2_ is depleted again.

The association of the high CO concentrations in the 65 °C thermal variant observed in this experiment with the activity of CODH-producing bacteria is based on the characteristics of the bacterial species capable of converting CO. The thermophilic CODH-producing strains discovered so far are more numerous than the mesophilic species [36]. The conditions prevailing in the temperature variant of 65 °C in the experiment carried out here were therefore optimal for a number of anaerobic bacterial strains for which the production of CODH was proven [37]. Oxygen deficit (2.7 L·h^−1^ aeration) together with high temperature in the climatic chamber could lead to the development of bacterial groups such as methanogenic, carboxydotrophic, hydrogenogenic, and acetogenic microorganisms, with potential representatives for which the optimum temperature is 65 °C: others *Methanothermobacter thermautotrophicus*, *Thermoanaerobacter kivui*, *Carboxydothermus pertinax*, *Carboxydothermus islandicus*, *Calderihabitans maritimus* KKC1, among others [37]. It has also been proven that with increasing temperature, CODH activity increases and is associated with a greater yield of CO_2_ to CO conversion [38] and, thus, in the high-temperature variant of this experiment, the expression of the CODH gene in the bacteria colonizing the composted waste could occur.

The biological production in this temperature variant may also be the reason why it was characterized by the greatest randomness, and the measured CO concentrations differed significantly from each other (high standard deviation for samples at 65 °C). Different bacterial strains, characterized by different efficiencies of CO production, could have appeared in individual reactors [39]. In turn, the analysis of the kinetics of the decrease in CO concentration during composting showed that although C_COmax_ was the highest for an aeration of 2.7 L·h^−1^ in each temperature variant, the reaction rate constant k for this level of aeration was highest at 55 °C, not 65 °C. This can be explained by the doubling time of CODH-producing bacteria, for example, for the *Methanothermobacter thermautotrophicus* strain, developing optimally at 65 °C, its doubling time is reported as extremely slow, reaching up to 200 h [37]. 

However, it is not only the thermophilic conditions in the experiment that indicate the biological production of CO during composting in oxygen-deficient areas. As proved by the analysis of variance, the lowest of the analyzed temperatures (35 °C) had a significant impact on the concentration of CO. Additionally, the correlation analysis showed that both for the 35 °C and 65 °C variants, the CO concentration was negatively correlated with the availability of oxygen, which can also be explained by the activity of CODH-producing anaerobes. In addition to species developing at temperatures >65 °C, this group also includes mesophiles, and the optimum point of their activity is in the thermal range of 30–37 °C. The strains known so far that function in the environment with CO include *Methanosarcina barkeri*, *Methanosarcina acetivorans*, *Alkalibaculum bacchi*, and *Butyribacterium methylotrophicum* (with an optimum reached at 37 °C) or *Acetobacterium woodii*, *Rhodospirillum rubrum* and *Clostridium drakei* (with an optimum at 30 °C) [37]. It should be emphasized that although the bacterial strains discussed above have been analyzed for CO conversion using CODH, there are no studies that analyze the ability of the same bacteria to carry out the reverse process: net CO production using the same enzyme. The release of a small amount of CO during laboratory analyses at the end of the 20th century, noted in the case of *Moorella thermoacetica* and *Methanothermobacter thermautotrophicus*, did not lead to the continuation of research [40].

Although the composting process in individual temperature conditions (35, 45, 55, 65 °C) was carried out separately, each of the analyzed variants takes place in a real, traditional composting process. Starting from the mesophilic phase, when the temperature of the material increases from 35 °C to 45 °C, the decomposition of organic matter in the composted waste generates heat and the pile or bioreactor becomes thermophilic (55–65 °C and above) [41]. Combining this information with the results obtained in this study, it can therefore be assumed that CO production under oxygen-deficit conditions follows changes in the microbial community in the waste, which are caused by process temperature phases. In this way, CO can be produced by CODH-producing bacteria; first by mesophilic species growing at 35 °C, and then by thermophiles as the process temperature increases (65 °C). Such a trend is consistent with the observations of other researchers who noted the second peak of CO production when the temperature of the material after previous cooling increased again to thermophilic conditions [7,16].

As mentioned earlier, waste composting is currently not seen as a technology with the potential to be coupled with biorefinery processes. However, the observations made during this research may lead to the formulation of recommendations for the composting process focused on CO production. According to the results obtained, CO production is significantly affected by low aeration (<3.4 L·h^−1^) and low temperature (<45 °C). From a practical point of view, composting aimed at generating a large net CO could therefore be carried out in economically effective conditions, based on the low efficiency of aeration systems. The composting process system would change; when controlling the thermal conditions in the pile or bioreactor, it would be advantageous to extend the mesophilic phase with temperatures close to 35 °C and not exceeding 45 °C. Then, in order to hygienize the material and simultaneously generate CO in thermophilic conditions, the material would be exposed to a high temperature (65 °C). Such an artificially imposed system, however, requires prior analysis in controlled laboratory conditions, and then during pilot composting processes on a semi-technical or technical scale in order to assess the quality of the final product of the process.

Additional requirements for a plant that utilizes bio-waste for the production of CO are of course strict safety considerations due to the toxicity of the gas and developments towards the capture and purification of CO. Although CO is a key reactant in many processes, which can lead to the production of many valuable chemical compounds, its use is limited by the need to obtain a high purity of gas stream [9]. Obtaining CO in a concentration exceeding 99 mol% therefore requires an energy-efficient separation process. Despite significant research in developing new CO separation technologies in recent years, this problem still remains unresolved, and developed methods such as cryogenic distillation, absorption, membrane or adsorptive separation still face a lot of challenges [9]. Among the weaknesses of these solutions, there are the unsolvable problems regarding CO and N_2_ separation based on the similar boiling points of these compounds during cryogenic distillation or low recovery rates with insufficient CO concentration feeds for adsorption. In turn, despite the elimination of these problems in the case of methods based on absorption, environmental and safety aspects appear, such as the disposal of spent and volatile solvents [9]. Due to the need of using CO separation for industrial purposes, the problem of high costs is not without significance. It can be based on capital expenditures, such as the purchase of a cooling utility for cryogenic distillation, as well as operating costs, including the use of pretreatment to remove impurities (adsorption and cryogenic distillation) or a multistage process to yield in higher purity (membrane separation) [42].

In the context of the discussed problem, it is also important to take into account the scale of the composting process and the feedstock available for the process. Only in the European Union does composting annually process 42 million tons of bio-waste (59% of the total stream). However, by 2035, the number of tons is projected to increase by another 40 million tons per year. Thus, from 3800 composting plants in 2022, the number of installations will increase to approx. 7600 [43]. Therefore, the existing and future infrastructure, as well as the increasing supply of substrates, are favorable conditions for the development of this niche in the circular economy.

## 5. Conclusions and Future Research Recommendations

Based on the findings of this study, suggestions for improving CO production throughout the composting process have been developed for the first time. The conducted research proved that the production of CO during bio-waste composting on a laboratory scale depends on the aeration rate and the process temperature. The highest concentrations of CO in each thermal variant was achieved with an oxygen deficit (aeration 2.7 L·h^−1^). Additionally, CO levels increased with temperature, reaching concentrations of ~300 ppm at 65 °C. The production of CO in mesophilic and thermophilic conditions (35 °C and 65 °C variants) highlights the biological nature of CO formation by microorganisms capable of producing the CODH enzyme. For this reason and taking into account the high standard deviations between CO concentrations in the variant with the highest process temperature, further research is necessary on the efficiency of obtaining CO in these thermal ranges, possibly in specially dedicated bioreactors. It is also necessary to characterize the microbial community involved in the process under these process conditions and to analyze the ability of the identified bacteria to produce the CODH enzyme, and to analyze the direction of CO and CO_2_ conversion.

## Figures and Tables

**Figure 1 materials-16-04551-f001:**
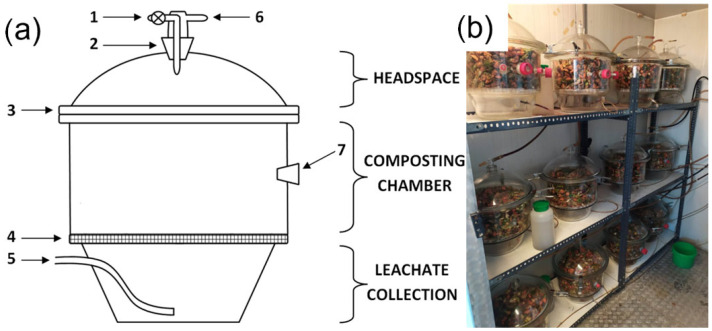
Bioreactors for laboratory-scale composting: (**a**) scheme of the bioreactor (based on [21]), 1—closed valve, 2—covers screw cup, 3—cover edge, 4—perforated plastic screen, 5—aeration hose, 6—exhaust air outlet, 7—screw cup for temperature measurements; (**b**) bioreactors in the climate chamber.

**Figure 2 materials-16-04551-f002:**
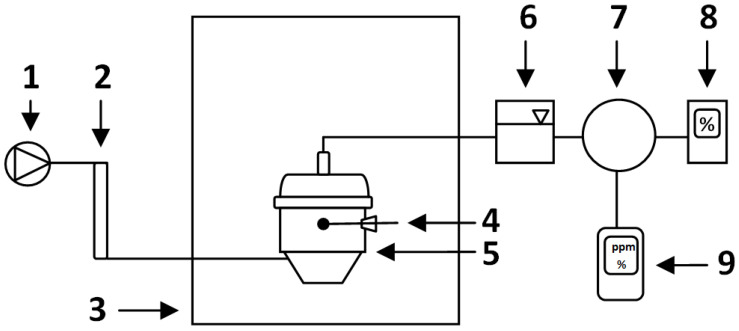
Laboratory-scale composting system (based on [21]): 1—air supply, 2—flowmeter, 3—climate chamber, 4—thermocouple, 5—bioreactor, 6—cooling system, 7—puffer bag, 8—IR gas analyzer (CO_2_, % *v*/*v*), 9—gas concentration analyzer (CO, ppm; O_2_, % *v*/*v*).

**Figure 3 materials-16-04551-f003:**
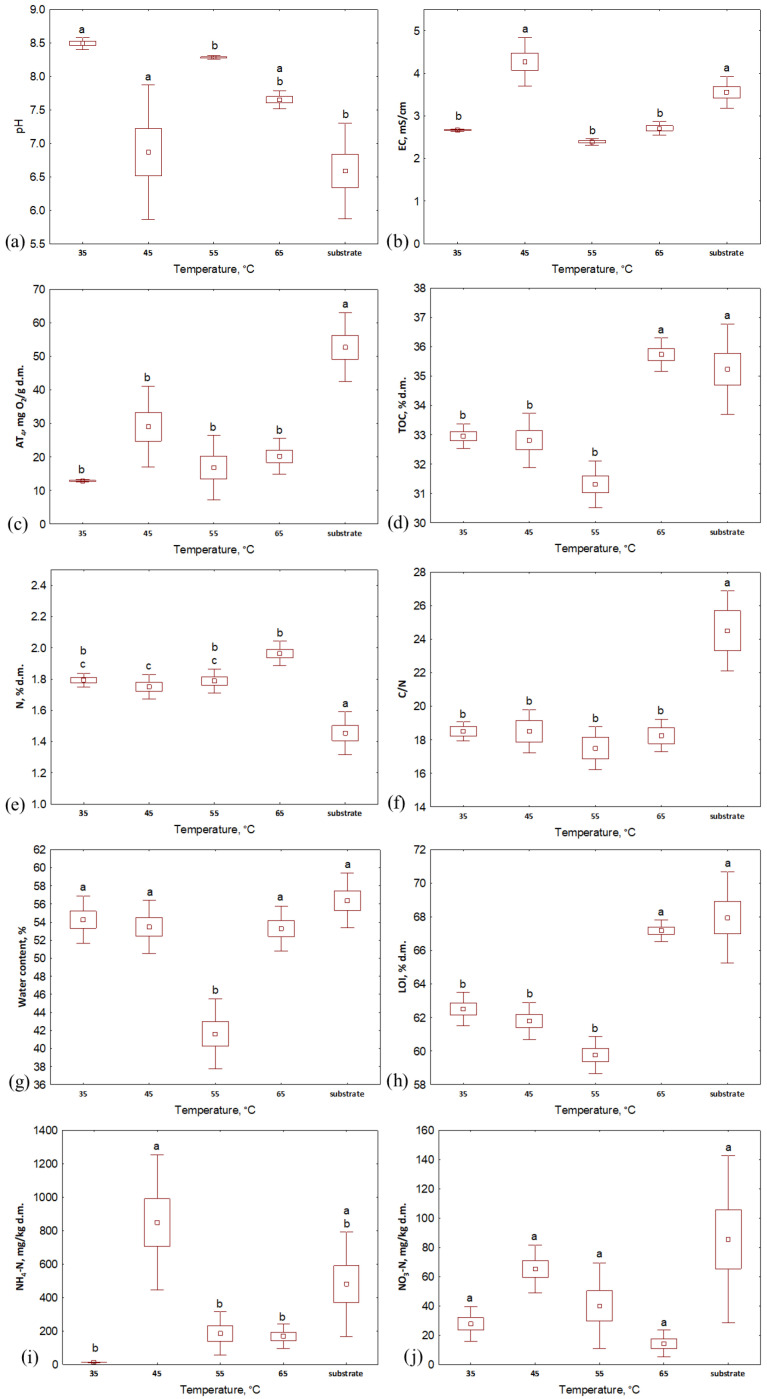
Properties of substrate and composts after 14 days of the composting process in different temperature variants: (**a**) pH; (**b**) EC, mS·cm^−1^; (**c**) AT_4_, mg O_2_·g d.m.^−1^; (**d**) TOC, % d.m.; (**e**) TN, % d.m.; (**f**) C/N; (**g**) water content, %; (**h**) LOI, % d.m.; (**i**) NH_4_-N, mg·kg d.m.^−1^; (**j**) NO_3_-N, mg·kg d.m.^−1^; letters (a, b, c) indicate the homogeneity group according to Tukey’s test at a significance level *p* < 0.05.

**Figure 4 materials-16-04551-f004:**
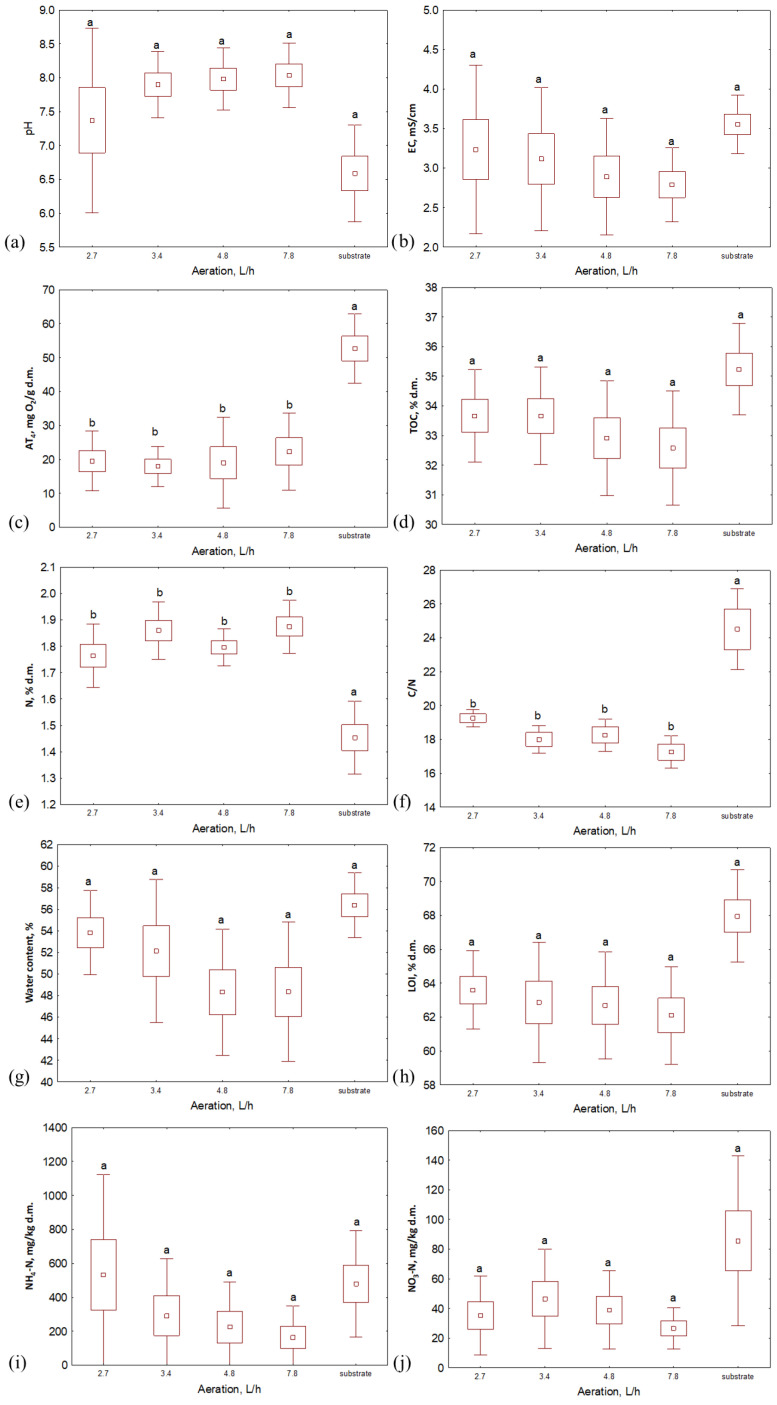
Properties of substrate and composts after 14 days of the composting process in different aeration rate variants: (**a**) pH; (**b**) EC, mS·cm^−1^; (**c**) AT_4_, mg O_2_·g d.m.^−1^; (**d**) TOC, % d.m.; (**e**) TN, % d.m.; (**f**) C/N; (**g**) water content, %; (**h**) LOI, % d.m.; (**i**) NH_4_-N, mg·kg d.m.^−1^; (**j**) NO_3_-N, mg·kg d.m.^−1^; letters (a, bindicate the homogeneity group according to Tukey’s test at a significance level *p* < 0.05.

**Figure 5 materials-16-04551-f005:**
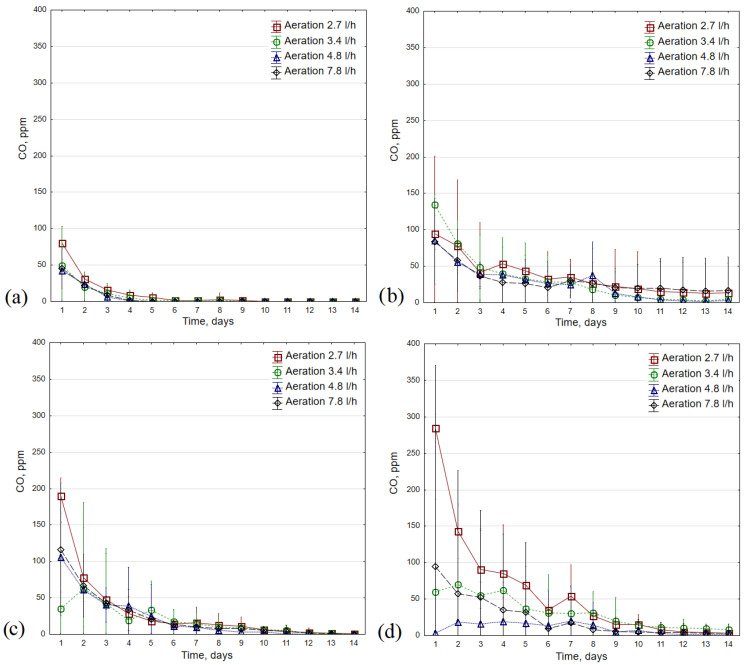
CO concentration average values (±standard deviation) during 14 days of the composting process in different temperature variants: (**a**) 35 °C; (**b**) 45 °C; (**c**) 55 °C; (**d**) 65 °C.

**Figure 6 materials-16-04551-f006:**
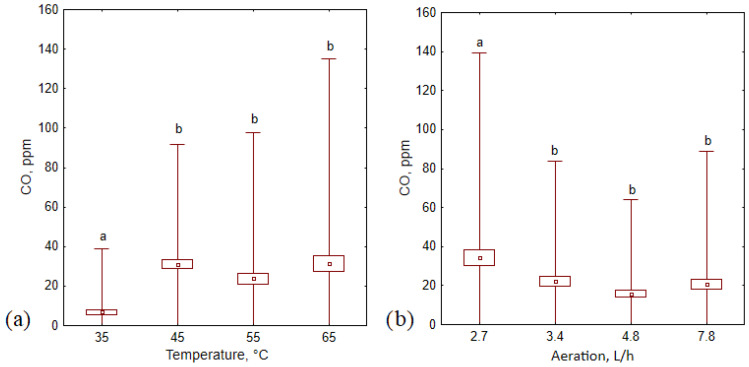
Average CO concentration (±standard deviation) during 14 days of the composting process in (**a**) different temperature variants, and (**b**) different aeration variants; letters (a, b) indicate the homogeneity group according to Tukey’s test at a significance level *p* < 0.05.

**Table 1 materials-16-04551-t001:** Experimental design for laboratory composting.

Composting Series #	Compost Substrates	Duration of the Process, Days	Temperature, °C	Aeration Rate, L·h^−1^
1	Bio-waste (green waste and vegetables) mixed with chopped branches	14	35	2.7
				3.4
				4.8
				7.8
2			45	2.7
				3.4
				4.8
				7.8
3			55	2.7
				3.4
				4.8
				7.8
4			65	2.7
				3.4
				4.8
				7.8

**Table 2 materials-16-04551-t002:** Properties of substrates for the composting process in different temperature variants (average ± std. dev.); d.m.—dry matter.

	Substrates for the Composting Process			
Properties ± std. dev.	35 °C	45 °C	55 °C	65 °C
pH	7.48 ± 0.07	5.60 ± 0.01	6.51 ± 0.11	6.75 ± 0.09
EC, mS·cm^−1^	3.53 ± 0.03	4.06 ± 0.05	3.09 ± 0.06	3.51 ± 0.04
TOC, % d.m.	35.66 ± 0.07	33.81 ± 0.47	33.03 ± 0.12	36.94 ± 0.83
TN, % d.m.	1.26 ± 0.02	1.50 ± 0.01	1.44 ± 0.03	1.62 ± 0.03
C/N	28	24	23	23
Water content, %	54.40 ± 0.30	55.70 ± 0.89	54.28 ± 0.32	61.00 ± 0.23
LOI, % d.m.	69.19 ± 0.26	67.48 ± 0.44	64.15 ± 0.30	71.01 ± 0.37
AT_4_, mg O_2_·g d.m.^−1^	37.6 ± 0.2	64.0 ± 2.8	61.7 ± 1.2	71.1 ± 10.0
NH_4_-N, mg·kg d.m.^−1^	140.46 ± 3.02	850.40 ± 13.76	689.28 ± 44.56	248.08 ± 1.63
NO_3_-N, mg·kg d.m.^−1^	58.22 ± 6.10	82.90 ± 6.36	30.42 ± 12.69	169.62 ± 26.65

**Table 3 materials-16-04551-t003:** Relative process efficiency in different temperature variants (%).

Process Temperature, °C	Aeration Rate, L·h^−1^	pH	EC	AT_4_	TOC	TN	C/N	Water Content	LOI	NH_4_-N	NO_3_-N
35	2.7	12.47	−24.54	−65.38	−6.17	38.93	−32.14	3.27	−8.21	−90.94	−51.23
	3.4	12.47	−25.25	−65.65	−6.87	43.77	−35.71	1.65	−8.89	−89.95	−33.10
	4.8	13.87	−24.54	−65.65	−8.56	39.05	−32.14	−5.27	−9.83	−92.11	−55.96
	7.2	15.08	−23.40	−66.18	−8.70	46.42	−35.71	−0.73	−11.70	−93.21	−66.07
45	2.7	−6.59	21.20	−78.99	−6.02	9.92	−16.67	1.01	−6.37	74.25	−12.06
	3.4	31.17	11.64	−51.58	−4.22	17.24	−20.83	0.59	−7.68	−2.08	−32.01
	4.8	32.06	−0.49	−27.23	−9.66	22.50	−29.17	−7.49	−10.22	−22.49	−4.45
	7.2	32.50	−13.60	−32.91	−8.29	18.12	−25.00	−10.23	−9.39	−45.47	−44.19
55	2.7	27.59	−19.68	−46.19	−3.02	19.07	−17.39	−12.36	−5.15	−43.29	−7.94
	3.4	27.44	−22.90	−76.23	−4.06	28.45	−26.09	−23.68	−8.93	−80.02	184.66
	4.8	26.83	−26.45	−85.02	−5.16	20.02	−21.74	−28.44	−6.15	−79.62	−21.03
	7.2	27.36	−22.90	−80.00	−8.54	29.52	−30.43	−28.84	−7.18	−89.53	−30.12
65	2.7	11.64	−18.97	−63.33	−2.55	20.23	−17.39	−9.44	−5.87	6.97	−94.06
	3.4	11.56	−17.83	−69.03	−2.71	25.04	−21.74	−8.80	−4.67	−21.63	−98.09
	4.8	14.68	−26.96	−75.93	−3.06	15.45	−17.39	−15.98	−4.84	−61.60	−85.16
	7.2	15.72	−27.25	−52.55	−4.80	24.82	−26.09	−16.96	−6.18	−53.74	−89.25

**Table 4 materials-16-04551-t004:** The bio-waste weight loss during composting under different process conditions.

Aeration Rate, L·h^−1^	Weight Loss, %	35 °C	45 °C	55 °C	65 °C
2.7	Total	11.44 ± 1.05	4.90 ± 2.50	7.48 ± 0.82	9.94 ± 0.96
	As leachate	5.95 ± 1.03	0.99 ± 1.71	16.47 ± 1.65	9.20 ± 2.98
3.4	Total	11.66 ± 1.16	8.09 ± 3.87	12.51 ± 0.85	12.04 ± 2.02
	As leachate	6.18 ± 0.35	2.88 ± 2.50	18.84 ± 2.47	5.41 ± 4.69
4.8	Total	13.64 ± 0.41	15.83 ± 9.10	16.15 ± 0.09	16.20 ± 0.37
	As leachate	5.32 ± 0.83	7.43 ± 7.21	20.03 ± 0.82	12.31 ± 1.18
7.8	Total	13.59 ± 0.76	12.88 ± 8.07	13.26 ± 1.27	16.14 ± 2.94
	As leachate	5.80 ± 1.86	4.71 ± 4.93	20.10 ± 2.88	14.61 ± 4.09

**Table 5 materials-16-04551-t005:** Correlation analysis between CO and CO_2_, O_2_ concentrations, ambient temperature, and compost temperature for thermal variants of the composting process (first 7 days); * shows values with Pearson r correlation coefficients that were statistically significant with *p* < 0.05.

	The Composting Variant, °C	Ambient Temperature, °C	Compost Temperature, °C	CO_2_, %	O_2_, %
CO, ppm	35	0.22 *	0.59 *	0.40 *	−0.47 *
	45	−0.03	−0.24 *	−0.03	0.09
	55	−0.11	−0.56 *	−0.53 *	0.55 *
	65	0.31 *	0.16	0.22 *	−0.11

**Table 6 materials-16-04551-t006:** CO production kinetics during composting under different temperatures and aeration rates.

Process T, °C	Aeration, L·h^−1^	Reaction Order	C_CO max_, ppm	k, d^−1^	a = k·C_CO max_, ppm·d^−1^
35	2.7	1st-order	185.3 ± 21.4	0.846 ± 0.025	156.5 ± 15.6
	3.4	1st-order	109.5 ± 52.5	0.822 ± 0.185	90.4 ± 43.1
	4.8	1st-order	101.0 ± 19.0	0.834 ± 0.160	86.2 ± 32.5
	7.8	1st-order	120.1 ± 77.7	0.850 ± 0.335	119.4 ± 120.5
45	2.7	1st-order	103.1 ± 44.0	0.185 ± 0.053	18.5 ± 8.8
	3.4	1st-order	214.6 ± 84.7	0.453 ± 0.301	114.2 ± 116.2
	4.8	1st-order	95.6 ± 12.4	0.218 ± 0.041	21.1 ± 6.5
	7.8	1st-order	80.2 ± 18.2	0.179 ± 0.092	14.9 ± 10.2
55	2.7	1st-order	471.9 ± 283.4	0.816 ± 0.496	478.8 ± 530.0
	3.4	1st-order	168.8 ± 84.4	0.338 ± 0.088	61.9 ± 46.1
	4.8	1st-order	153.0 ± 19.3	0.410 ± 0.051	62.2 ± 5.0
	7.8	1st-order	183.9 ± 87.3	0.434 ± 0.157	89.0 ± 61.6
65	2.7	1st-order	403.6 ± 43.1	0.422 ± 0.090	172.8 ± 56.2
	3.4	1st-order	127.5 ± 89.1	0.183 ± 0.111	29.9 ± 27.6
	4.8	0th-order	–	–	0.09 ± 0.1
	7.8	1st-order	161.9 ± 152.2	0.670 ± 0.620	92.0 ± 73.2

## Data Availability

The data presented in this study are contained within the article and its supplementary material.

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
