# Peer review of "Carbon Monoxide Production during Bio-Waste Composting under Different Temperature and Aeration Regimes"

_materials, 2023, doi:10.3390/ma16134551_

Round 1
Reviewer 1 Report
After reading this manuscript, major comments are drawn below:
1 - Please, revise ranging temperature in the document - "psychrophilic to thermophilic (T=35, 45, 55, and 65°C)". I think it means mesophilic and thermophilic temperature.
2 - Implications of the study's findings for the existing literature are missing in abstract and conclusions.
3 - Introduction: add the study's novelty compared with other published in the scientific literature.
4 - Materials and methods: Why were these experimental conditions (i.e., aeration and temperature) defined?
5 - Tables and figures should be self-explaining. Please, identify acronyms at the end of each of them or add a list of abbreviations.
6 - lines 494-510: What would be the technological application for CO recovered from composting? Please, elaborate more on this aspect.
7 - Overall, it is unclear why recovering CO from composting process would have a practical and technological application. Optimized composting aims to mineralize organic matter to CO2 and H2O. Why would be CO recovering attractive? and, would be its concentration significant? Authors are advised to clarify this aspect of the manuscript.
Author Response
All responses to reviewer comments are in the attached file.

Reviewer 2 Report
- The subject of this study is very interesting, especially in our time, where ecological aspects are more important than ever. Nevertheless, it is purely experimental, I wonder if it is possible to do numerical simulations to simulate this process in order to save time and resources.
- The results of figure 3 are very interesting but the figure is not clear for the reading. I propose to improve it, or perhaps, to put it in the form of several figures and to enlarge them to better read clearly.
- The same remark for figure 4.
- Table 1 is too big, it needs to be reduced a bit.
- The same remark for figure 2.
- Add some references related to this work: DOI10.1177/00219983221138951 and DOI10.3390/su14148638 and
Author Response

(The authors gave the same response as above.)

Reviewer 3 Report
The manuscript is devoted to the study of the process of obtaining carbon monoxide by composting model biowaste. This approach is certainly of interest in connection with the possibility of extracting valuable products from biowaste.
However, the practical implementation of this scientific topic is likely to be limited by low concentrations of carbon monoxide and, consequently, the need to separate carbon monoxide from other gases. Is there an assessment of the practical applicability of this direction in the scientific literature? This aspect should be presented in more detail in the introduction, and not just in the discussion. Are there any approaches to separating and accumulating such low oxide concentrations?
The complete composition of the gas mixture is not presented. Are other gaseous oxides registered besides CO and CO2? Is it possible to make an assumption about what is the source of CO, the concentration of which gradually decreases as a result of the aeration process?
Minor editing of English language required
Author Response

(The authors gave the same response as above.)

Round 2
Reviewer 1 Report
Minor comments:
“In the context of the discussed problem, it is also important to take into account the scale of the composting process and the feedstock available for the process. Only in the European Union, composting annually processes 42 million tonnes of bio-waste (59% of the total stream). However, by 2035, the number of tonnes is projected to increase by another 40 million tonnes per year. Thus, from 3,800 composting plants in 2022, the number of installations will increase to approx. 7,600 [38]. Therefore, the existing and future infrastructure, as well as the increasing supply of substrates, are favorable conditions for the development of this niche of the circular economy.”
38. Zhang, L.; Can, M.; Ragsdale, S.W.; Armstrong, F.A. Fast and Selective Photoreduction of CO2 to CO Catalyzed 698 by a Complex of Carbon Monoxide Dehydrogenase, TiO2, and Ag Nanoclusters. ACS Catal. 2018, 8, 2789–2795, 699 doi:10.1021/acscatal.7b04308
The presented data are not available in the cited reference. Authors are advised to check it and be more careful with the shared information. From where these data came?
"Despite significant research in developing new CO separation technologies in recent years, this problem still remains unresolved, and developed methods such as membrane separations or using the adsorbents and membranes for CO2 removal still face a lot of challenges [9].”
Please, identify these challenges and elaborate more on.
Author Response
The responses to the reviewer's comments are in the attached file.
